# Diet Quality and Contextual Factors Influencing Food Choice among Adolescents with Food Security and Food Insecurity in Baltimore City

**DOI:** 10.3390/nu14214573

**Published:** 2022-10-31

**Authors:** Kaitlyn Harper, Laura E. Caulfield, Stacy V. Lu, Kristin Mmari, Susan M. Gross

**Affiliations:** 1Department of Environmental Health and Engineering, Johns Hopkins Bloomberg School of Public Health, 615 N Wolfe Street, Baltimore, MD 21205, USA; 2Department of International Health, Johns Hopkins Bloomberg School of Public Health, 615 N Wolfe Street, Baltimore, MD 21205, USA; 3Department of Population, Family, and Reproductive Health, Johns Hopkins Bloomberg School of Public Health, 615 N Wolfe Street, Baltimore, MD 21205, USA

**Keywords:** food insecurity, nutrition insecurity, adolescents, dietary intake, dietary behaviors

## Abstract

This study evaluated differences in overall diet quality, diet quality components, and food-related contextual factors between adolescents with food security and those with food insecurity. Mixed methods analysis was conducted on data from three 24-h dietary recalls from 61 adolescents ages 14–19 years old living in Baltimore, Maryland, USA in 2020–2021. All adolescents were sampled from households eligible for the Supplemental Nutrition Assistance Program in 2020. There were no significant differences in overall diet quality or components between adolescents with food security and those with food insecurity in this sample, except for seafood and plant proteins, which was higher for adolescents with food insecurity. Qualitative analysis found that adolescents were largely influenced by their parents and the home food environment, and that workplace environments enabled adolescents to eat foods high in refined grains, sugar, and saturated fat. These findings provide insight about the experiences of low-income adolescents during times when they are home for prolonged periods (i.e., emergency school closures, summer, and winter breaks). Programs and policies that aim to improve healthy food access may positively impact adolescent food security and diet quality, and it is important to ensure that healthy foods are available and accessible to adolescents in the places where they spend the most time. Multilevel interventions in the home, school, and workplace may be most effective in encouraging healthy eating behaviors among adolescents.

## 1. Introduction

In the United States, 14.8% of households with children experience food insecurity, defined as having inconsistent access to adequate food, primarily due to a household’s lack of economic resources [1]. Food insecurity is typically measured at the household level using a series of questions that ask about the frequency of experiencing difficulty purchasing food specifically due to lack of money. The questions increase in severity, beginning with the frequency of worrying that food will run out and ending with skipping meals for an entire day [2]. In general, food security modules do not inquire about the healthfulness of foods eaten in the household, but some versions include questions about members of the households’ consumption of “cheap foods” and ability to eat “balanced meals”. Although the definition and measurement of food insecurity do not focus on the quality of foods consumed, it is well-documented that food insecurity is associated with lower diet quality in adults [3,4,5,6,7,8,9]. This association may be attributable to differences in cost and availability of “healthy” versus “unhealthy” foods—i.e., healthy foods (with higher nutrient density) are associated with higher cost per kcal whereas unhealthy foods (with higher energy density and lower nutrient density) are associated with lower cost [10,11]. Therefore, individuals experiencing food insecurity may have limited ability to purchase and consume healthy foods.

With this logic, it would be reasonable to assume that children living in food-insecure households would also consume less healthy foods. However, studies that have evaluated the association between diet quality or components of diet quality and food insecurity among children report mixed results. For example, in their systematic review, Hanson and Connor (2014) found that only 16% of 130 associations tested in populations of children ages 2–19 had significant associations between food insecurity and poor diet quality measures; 3% showed a beneficial effect on diet quality measures, and 81% showed either ambiguous, inconsequential, or null effects [3]. One explanation is that parents may shield their children from the effects of food insecurity, which may improve the types of foods accessible to and consumed by children. Younger children may be shielded more than adolescents, resulting in greater risk of poor diet quality for adolescents experiencing food insecurity [12,13]. Additionally, the Special Supplemental Nutrition Program for Women, Infants, and Children (WIC) provides benefits which support better diet quality for young children in low-income households, but no such federal program specifically targets older children or adolescents. The school meals programs (e.g., School Breakfast Program, National School Lunch Program, Summer Food Service Program) provide free- and reduced-price nutritious meals to school-aged children and adolescents in low-income households [14,15]; however, adolescents have reported feeling disincentivized to use school meals programs due to stigmatization [16].

Few studies have assessed the association between food insecurity and nutrition specifically in adolescents; and those that exist show inconsistent results. Additionally, most studies have used limited metrics to assess individual foods (e.g., fruits and vegetables, packaged snacks, sugar-sweetened beverages) or nutrients (e.g., iron, calcium, vitamin C) consumed by adolescents [3]. These measures evaluate only one aspect of the diet and, when assessed in isolation, may not provide a complete or accurate picture of overall diet quality. For instance, data from the National Health and Nutrition Examination Survey (NHANES) found that household food insecurity was associated with lower iron intake in all adolescents but was only associated with lower calcium intake in younger adolescent males (the association was not found in females or in a wider age range of males) [17,18]. Another study of NHANES data found an association with lower Healthy Eating Index (HEI) scores in a population of adolescents ages 15–17 years but did not assess younger or older adolescents [19]. A recent cross-sectional study of adolescents in Minnesota found that food insecurity was associated with reduced odds of consuming recommended amounts of vegetables, fruits, and milk per day [20]. In contrast, a large cohort study in Minnesota found that adolescents with food insecurity were more likely to eat at least three servings of vegetables with at least one serving of yellow or green vegetables, although they also had higher fat intake and were more likely to eat fast food compared to adolescents with food security [21].

Food choice plays a large role in determining diet quality, and adolescents experience more autonomy in their ability to choose their own foods compared to younger children. Individual psychosocial factors (e.g., beliefs, knowledge, self-efficacy, food preferences), biological factors (e.g., hunger, mental health), behavioral factors (e.g., timing of eating events, dieting behaviors), and lifestyle factors (e.g., food costs, time demands, physical activity) are related to food choice among adolescents [22]. Additionally, contextual factors include social and physical environmental influences, such as at school and in the household [22]. For example, studies have shown that peers and peer pressure play a large role in food-related decision-making among adolescents, particularly in school settings [23,24]. Other factors at school, such as convenience of options, timing of lunch schedules, and food quality and taste play a role in adolescents’ food choice [25]. Multiple systematic reviews have shown that adolescent consumption of fruits and vegetables is influenced by parental behaviors and intake [26,27,28], and studies have found that diet quality is positively associated with family income and parental education level [29,30]. Healthy food availability in the home has also been associated with healthy food consumption among US adolescents [31].

Despite the rich body of literature focusing on food choice among adolescents, previous studies have not considered food security status as a potential factor. It is important to examine factors that affect food choice and diet quality among adolescents with food insecurity and how they may be different from factors affecting adolescents with food security. Therefore, the aims of this study were to evaluate the association of food security status with overall diet quality and its components and to examine factors influencing food choice for adolescents with food security and those with food insecurity.

## 2. Materials and Methods

### 2.1. Study Design

This study was part of a larger evaluation of how federal nutrition assistance programs affect the food security status of adolescents living in Baltimore, Maryland during the COVID-19 pandemic. Briefly, the sample for this study included 64 individuals drawn from the larger survey sample of 281 adolescents ages 14–19 years who completed an application to YouthWorks, a Baltimore City youth employment program, and who lived in households eligible to receive Supplemental Nutrition Assistance Program (SNAP) benefits. Surveys were completed between October 2020 and January 2021, and 24-h dietary recall data for this sub-study were collected between December 2020 and March 2021.

This study utilized a sequential explanatory mixed methods study design. Mixed methods is a procedure for collecting, analyzing, and integrating both quantitative and qualitative data throughout the research process to gain a better understanding of the research problem [32]. Sequential explanatory mixed methods study designs consist of two phases—quantitative followed by qualitative, and the qualitative data are used specifically to explain or elaborate on quantitative results [33]. In the present study, quantitative and qualitative data were collected concurrently, but quantitative data were analyzed prior to qualitative analysis. After separate analyses, the quantitative and qualitative results were integrated and compared, and qualitative data were used to inform interpretation of the results of quantitative data.

Information about the potential opportunity to participate in this sub-study was included in the consent process of the survey. Participants over 18 years provided oral consent before completing the survey. For participants under 18 years, researchers called parents or legal guardians to obtain oral consent and subsequently obtained oral assent from the adolescents before completing the survey.

### 2.2. Setting

Data collection for this study took place in Baltimore during the first year of the COVID-19 pandemic. Restrictions related to COVID-19 (e.g., lockdowns, mask mandates) varied by state in the US. In Maryland at the time of data collection, individuals were not required to stay in their houses, but numerous indoor (e.g., schools, sports and music venues) and outdoor spaces (e.g., parks, sports fields and courts, beaches, dog parks, and playgrounds) were closed, large gatherings of people were prohibited, face coverings were mandated in all public spaces, and social distancing was encouraged. During this time, most children and adolescents attended online school from home. Free meals were distributed through the US federally funded Summer Food Service Program at numerous sites, such as schools, libraries, and recreation centers, for families with children under the age of 18.

### 2.3. Recruitment and Data Collection

For this sub-study, 64 participants were purposively sampled by age, gender, and food security status from the larger sample of survey respondents. Age was categorized into three groups: younger (14–15 years), mid (16–17 years), and older (18–19 years). Gender was dichotomized into two groups, boys and girls plus non-binary individuals, due to the small sample size of non-binary participants. Food security was measured using a 9-item tool, created in 2004 by Connell and colleagues to assess food security status in older children at the individual level rather than the household level [34]. Respondents were sampled in two categories based on the number of affirmative responses to the 9-item tool: food secure (0–1) and food insecure (2–9).

Twenty-four-hour (24 h) dietary recalls were completed with each participant using the Automated Self-Administered 24-Hour Recall Dietary Assessment Tool (ASA24) version 2021 (https://epi.grants.cancer.gov/asa24/, 15 December 2020), developed by the National Cancer Institute, Bethesda, MD, USA. The target number of recalls for each participant was three; however, participants were included in the study if they completed at least two recalls. Participants were contacted three times over text and phone by researchers before considered lost to follow up.

During each recall, participants were asked to recount all foods and drinks consumed within the past 24 h. Additionally, for each food, participants were asked to recall information about food acquisition and eating behaviors, including what time the food was eaten, where the food was obtained, who purchased and/or prepared the food, what they were doing while they ate, and who they ate with. These data were recorded on a Food Log Spreadsheet created in Excel (Figure A1). During each recall researchers wrote memos with contextual information about each participant (e.g., family members, availability of transportation, whether they attended school, whether they had a job). To obtain contextual information, structured interview questions were asked at the beginning of each recall (Appendix B). Recalls lasted approximately 30 min each and were conducted approximately one week apart, including two weekdays (Monday through Friday) and one weekend day (Saturday and Sunday) when possible. Participants received a $20 gift card after completing each recall.

### 2.4. Measures Used in These Analyses

#### 2.4.1. Dietary Measures

The ASA24 software calculates foods, food types, nutrients, and number of meals for each recall. Healthy Eating Index 2015 (HEI-2015) scores were calculated from the ASA24 output. HEI-2015 is a measure of how well an individual’s diet aligns with the Dietary Guidelines for Americans [35]. It is scored from 0 (lowest) to 100 (highest) points and calculated using 13 equally weighted components that include nine “adequacy components” (total fruits, whole fruits, total vegetables, greens and beans, whole grains, dairy, total protein foods, seafood and plant proteins, and the ratio of fatty acids to saturated fats) and four “moderation components” (refined grains, sodium, added sugars, and saturated fats) [36]. Higher points are awarded for higher consumption of adequacy components and for lower consumption of moderation components. Maximum scores for total fruits, whole fruits, total vegetables, greens and beans, total protein foods, and seafood and plant proteins are 5 points; maximum scores for all other components are 10 points. We calculated HEI-2015 total and component scores using the simple HEI scoring algorithm from the publicly available SAS macros available on the National Cancer Institute website [37].

#### 2.4.2. Non-Dietary Measures

Demographic characteristics were collected in the survey and used to describe the sample, including race (Black, Multiracial), adolescent employment (no job, part-time, full-time), housing stability (dichotomized based on whether the respondent had slept in multiple locations within the past month due to economic instability/safety concerns), and number of people in the household (dichotomized into 1–4 people or more than 4 people). Most survey participants identified as Black/African American, which is consistent with the demographic distribution for YouthWorks applicants (Email conversation, Baltimore City Mayor’s Office of Employment, 22 October 2021).

### 2.5. Analysis

#### 2.5.1. Preliminary Analyses

One participant was excluded from analysis due to poor data quality and two were excluded due to small sample sizes in each gender category (non-binary and gender not listed). A total of 61 participants were included in the analysis. Statistics were completed using R Statistical Program (Vienna, Australia, 2020) and *p*-values less than 0.05 were considered statistically significant. Descriptive statistics were used to explore demographic variables for the total sample and by food security status.

#### 2.5.2. Diet Quality

Univariate and bivariate frequencies were used to explore dietary and non-dietary variables. Bivariate frequencies were calculated for each non-dietary variable by food security status. The difference in medians between food-secure and food-insecure groups was calculated, as well as the association between food security status and HEI-2015 dietary component and overall HEI-2015 scores. Due to non-normality of data, the Kruskal–Wallis test was used to compare differences in dietary outcomes by food security status.

#### 2.5.3. Eating Behaviors

Content analysis was used to analyze food acquisition and food behaviors of participants. A similar technique was used previously to describe dietary patterns of low-income women [38]. First, case-based descriptive summaries were written based on recall memos and data recorded in the Food Log Spreadsheet. Summaries included patterns of food acquisition, food preparation, and eating behaviors at multiple nested levels: (i) between and across days; (ii) within each specific 24-h period; (iii) within each eating event; and (iv) within each food, including ingredients/components. Information from the memos was used to contextualize eating events on each day. For example, we noted patterns on days when participants were in school (either in-person or virtually), at work, or neither. We also looked for patterns when participants were by themselves or with siblings versus when parents/grandparents were present. Summaries were then analyzed using an iterative constant comparative process to identify themes [39]. Summaries from the food-secure and food-insecure groups were analyzed separately first to identify themes within each group, then together to identify differences. Memo writing was used to assist the researchers with reflexivity in the coding and analytic process.

In this analysis, meals eaten within two hours of waking up were categorized as “breakfast”. Breakfast is traditionally viewed as a morning meal; however, in this analysis, timing of breakfast was dependent on when the participant woke up. Foods consumed in the evening (past 5 pm) were categorized as “dinner”, regardless of the type of foods eaten during the meal. For example, one participant consumed only a large piece of cake in the evening, and this was written in the summary as having cake for dinner. Additionally, at times there were multiple adults involved in participants’ meal events. Most of the time parents or other guardians (e.g., grandparents, aunts) were involved, although occasionally extended family friends’ parents were involved. For brevity in this paper, these adults were all categorized under the umbrella term “parent”.

## 3. Results

### 3.1. Participant Characteristics

One hundred and eighty-two recalls (564 total eating events) were collected from the 61 participants, including 57 recalls (30.9%) on weekend days. Due to scheduling challenges, 3 of the 61 participants (4.9%) completed all three recalls on weekdays, and 5 participants (8.1%) completed two recalls on weekend days and one recall on a weekday. Five participants (8.2%) completed only two recalls. By design, approximately half of the participants in the study were classified as food insecure (Table 1). The sociodemographic profile of adolescents was generally similar between the two groups. However, slightly more food-secure participants lived in single-parent households, and slightly more food-insecure participants lived in households with more than four people. All but two participants attended virtual school from home in Baltimore, either through a high school (*n* = 48), community college (*n* = 4), or university (*n* = 7). Of the two that did not attend virtual school, one had previously graduated from high school and was not enrolled in college, and the other attended a private high school that met in-person during the pandemic.

### 3.2. Differences in Diet Quality between Adolescents with Food Security and Food Insecurity

The median total HEI-2015 score for all 61 participants was 46.5 (range: 27.1–66.7 out of 100 possible points). Overall, participants scored highest in the added sugars and refined grains categories (7.2 and 7.0, respectively, out of 10 possible points), indicating relatively *low* consumption of foods/beverages with added sugars and refined grains. Conversely, participants scored lowest in the whole grains and whole fruits categories (0.1 and 0.1, out of 5 possible points), indicating relatively low consumption of these food items. Notably, participants scored the maximum score (5 out of 5 possible points) for the total protein category, indicating high protein consumption among this population.

The median total HEI-2015 scores were 46.8 and 45.8 for adolescents with food security and those with food insecurity, respectively (Table 2). There were no significant differences between adolescents with food security and those with food insecurity for total scores or individual dietary components, except for seafood and plant protein, in which the median score for adolescents with food insecurity was 2.71 points higher than the median score for adolescents with food security (*p* = 0.02). Adolescents with food security scored slightly higher than adolescents with food insecurity for refined grains, added sugars, dairy, total vegetables, and saturated fats, but they were not statistically different. Adolescents with food insecurity scored slightly higher than adolescents with food security for fatty acids, total fruits, beans and greens, and whole grains, but again, the component scores were not statistically different.

### 3.3. Factors Influencing Diet Quality

As shown, diet quality was low among all adolescents in the sample, and diet quality components were largely similar between participants with food security and those with food insecurity. In this section, we explore factors that may have influenced food choices leading to higher or lower diet quality among all adolescents in the sample.

#### 3.3.1. Availability and Accessibility of Seafood and Peanut Butter at Home

Forty-four percent of participants with food insecurity reported consuming seafood and plant proteins compared to 29% of participants with food security. Although participants with food insecurity consumed seafood, including salmon, white fish, shrimp, and crab, more than participants with food security, factors influencing seafood consumption did not differ by food security status. Seafood was often available as a food choice at home in both groups and was most often prepared by a parent. Typically, seafood was part of a larger dinner meal containing multiple components. For example, seafood dishes were often served with one or more vegetable sides (e.g., cooked corn and spinach) or were part of combination meals such as fried rice, pasta, and stir fry. Peanut butter was the only source of plant protein consumed by participants; no other plant proteins (e.g., tofu, nuts, or pulses) were consumed. Notably, peanut butter was more frequently consumed by those with food insecurity than those with food security (19% and 3%, respectively). Participants reported preparing simple and convenient meals with peanut butter (e.g., adding peanut butter to oatmeal, making peanut butter sandwiches) or eating packaged sandwiches previously purchased by their parents from the grocery store.

#### 3.3.2. Parental Influence over Fruit and Vegetable Consumption

Overall, fruit and vegetable consumption was low in both food-secure and food-insecure groups. For example, 18 (58%) participants with food security and 15 (47%) participants with food insecurity reported consuming no fruit or fruit juice, and 9 (29%) participants with food security and 11 (34%) participants with food insecurity reported eating no vegetables over the three recalls. Those who reported consuming fruits and vegetables often did so because of parental influence. For example, participants in both groups reported that fruits and 100% juices were typically purchased by parents from the grocery store and kept in the house for the family. Additionally, most participants who ate vegetables reported eating them during dinners prepared by a parent, and participants with food security consumed vegetables prepared by parents more often than those with food insecurity (71% vs. 50%, respectively). Parents prepared a wide variety of vegetables, including side dishes of steamed broccoli, cooked corn, yams, collard greens or spinach, or vegetables in combination dishes, such as soup, lasagna, or spaghetti. Parents often ordered carryout when they did not prepare dinner; over one-third of participants with food security and food insecurity reported eating carryout for dinner at least one time over the three recalls. Vegetables were less often consumed from carryout or prepared by the participant themselves. However, three participants with food security and four with food insecurity reported preparing vegetables and did so in a variety of ways. For example, three participants ate bagged salad, one participant made steamed broccoli in the microwave, one prepared a spinach omelet, one added premade kimchi to her ramen, and one participant prepared multiple meals, including shrimp stir fry and two sandwiches with multiple types of vegetables. Notably, six participants with food insecurity reported skipping dinner at least one time. These six participants reported no vegetable consumption on the days when they did not eat dinner. Additionally, two other participants with food insecurity reported not eating dinner and no fruits or vegetables on any of the three recall days.

#### 3.3.3. Convenient Access to Unhealthy Snacks and Breakfast Foods at Home

All participants consumed added sugar and refined grains mostly in the form of packaged pastries (e.g., honeybun, donut), candy, fruit snacks, and sugar-sweetened beverages (e.g., sweet tea, soda). Most often, these items were either purchased by parents previously or purchased by participants at corner stores near their houses or at fast-food restaurants. Some participants described keeping items in their room to prevent others in the house from consuming them. A few participants in both the food-secure and food-insecure groups described consuming relatively large amounts of packaged snacks at one time. For example, one 18-year-old boy with food insecurity reported eating “30 Jolly Ranchers, one full-size chocolate bar, two Oreo brownies, four mini-size chocolate bars, and 35 cookie bites” during one eating event. Likewise, a 14-year-old boy with food security reported eating “Nine chocolate chip cookies, nine oatmeal cookies, a honeybun, a small bag of Goldfish, 5 cereal bars, and a bag of hot Cheetos”.

Breakfast was another source of added sugar and refined grains for both food-secure and food-insecure participants. Participants with food insecurity reported eating breakfast more consistently than those with food security and reported more often consuming simple meals for breakfast (e.g., easy-to-prepare foods, such as cereal or frozen waffles, or grab-and-go foods, such as granola bars, fruit, or donuts). Although some participants with food security also consumed simple meals, they reported eating meals that required preparation beyond the use of a microwave or toaster (e.g., eggs and sausage, smoothies) more often than participants with food insecurity. These more complex meals were often prepared by parents or siblings.

#### 3.3.4. Food Environment at Adolescent Workplaces

Approximately one-third of the participants were employed (Table 1). With one exception, participants with food security and those with food insecurity who worked at restaurants or other businesses that sell food (e.g., gas station) received free food or purchased food at discounted prices from their workplaces. Those who worked at non-food retail businesses ate out on the days they worked, buying food from nearby fast-food restaurants. Only one participant, a 19-year-old girl with food security who was a lifeguard, brought food from home to eat at lunch. Participants reported buying their own meals on workdays with the money they earned from working. Meals consumed on workdays often included energy-dense food items such as fried chicken, burgers, and fries, and never included vegetables other than potatoes (french fries) or lettuce on sandwiches. Participants most often ate during their break at work, although a few reported bringing food to eat at home after their shift. These foods were not shared with other members of the household, except for one 16-year-old boy with food insecurity, who explained that his parents frequently asked him to buy food from his job for members of the household.

## 4. Discussion

This study aimed to describe differences in overall diet quality, diet quality components, and factors influencing food choice between adolescents with food security and those with food insecurity. In our entire sample of 61 adolescents, HEI-2015 overall score was low (approximately 46.5 out of 100 possible points), which is consistent with national data. Average HEI-2015 scores for US adolescents ages 12–17 years range between 41 and 55, which are lower than the adult national average of 59 [40,41]. We found no significant differences in overall HEI-2015 score by adolescent-reported food security status. Previous studies have found mixed results between the association of food insecurity with various measures of diet quality in children and adolescents [3]. Our findings are inconsistent with a previous study of 2005–2008 NHANES data that assessed differences in HEI-2005 score by adolescent-reported food security status in adolescents ages 15 to 17 years, which found that adolescents with food security scored approximately six points higher than those with food insecurity (47 and 41 points, respectively) [19]. Notably, that study was conducted using data prior to implementation of the Healthy Hunger Free Kids Act in 2012, which would require schools to offer more fruits, vegetables, and whole grains and limit sodium, calories, and unsaturated fats in school meals. This could at least partially account for the differences in diet quality observed between adolescents with food security and those with food insecurity, as those with food insecurity may utilize school meals programs more often than those with food security. However, this may not apply to our study as none our respondents reported acquiring free meals from school.

What do we expect to see in terms of differences in HEI scores between adolescents with food security and those with food insecurity? To answer this, we must take a closer look at the questions used to measure adolescent-reported food insecurity (Table A1). Of the nine questions used to assess food insecurity, only two—questions 3 and 4—refer to differences in the types or quality of foods consumed by the adolescent because of their household’s inability to purchase food. For instance, question 1 refers to worry about food “running out”, but does not consider actual changes to the amount or types of foods in the household; question 2 refers to food “running out” and the household’s inability to purchase more, but it does not consider other food items that may be present in the household and that were not purchased (e.g., from food pantries); questions 5–9 refer to how much food consumed by individuals may differ as a result of eating less, cutting the size of meals, or skipping meals, but they do not consider differences in the types of foods consumed. Because HEI scores are calculated by density (e.g., amount per 1000 kcal, ratio of fatty acids), they are not affected by the size or number of meals consumed in a day. For example, if a food-insecure adolescent consumes a small number of calories consisting of mostly nutrient-dense foods, their HEI score may be higher than a food-secure adolescent who consumes a greater number of calories consisting of mostly energy-dense, nutrient-poor foods. In summary, seven of the nine questions asked on the adolescent-reported food security scale do not measure factors related to diet quality. Therefore, it is reasonable that there may not be differences in diet quality by adolescent food security status as measured by this survey. This may at least partially explain why we did not observe differences in HEI scores by food security status in this study.

Recently, there has been a shift in discourse from considering food security to considering nutrition security, defined by the USDA as having “consistent and equitable access to healthy, safe, affordable foods essential to optimal health and well-being” [42]. Compared to food security, nutrition security has a specific focus on the quality of foods. Therefore, measures of nutrition security may more accurately align with diet quality scores. Although some tools to measure nutrition security have been developed [43], these measures have yet to be tested for validity in adult, adolescent, or child populations.

The qualitative results of our study allowed us to explore and understand factors that influenced both higher and lower diet quality among all adolescents. For example, family played a prominent role in consumption of fruits and vegetables. Regardless of food security status, participants with higher diet quality scores reported eating fresh fruits and 100% fruit juices when their parents purchased them and kept them in the house. Additionally, most participants in our study ate vegetables only when parents prepared dinner for them. This is consistent with numerous previous studies that have found that adolescent consumption of fruits and vegetables is influenced by parental behaviors [26,27,28] and family meals [44,45,46,47].

Participants in our study who were employed also had consistent access to foods high in refined grains, sugar, and saturated fat through their workplace food environments. To our knowledge, no studies have explored the workplace food environment for adolescents or the impact it may have on diet quality. However, a systematic review identified limited availability of healthy foods, proximity of unhealthy foods, and limited facilities to prepare healthy food as factors influencing eating behaviors for adults in the workplace [48]. In 2019, approximately one-third of US adolescents ages 16–19 years were employed either part-time or full-time [49], and future research is needed on potential interventions that may encourage healthier eating for adolescents at their workplaces.

Circumstances related to the COVID-19 pandemic likely also impacted food choice and diet quality for adolescents. At the time of data collection, adolescents in this study were attending virtual school from home. During in-person school times, children and adolescents have access to nutritious meals through the National School Lunch Program and School Breakfast Program, which follow strict nutrition guidelines mandated by the United States Department of Agriculture. Additionally, in Baltimore City, all students have the ability to receive free breakfast and lunch at school through the Community Eligibility Provision, which allows schools and school districts to provide free meals if over 40% of the student population lives in SNAP-eligible households [50]. Although numerous school meal sites offered grab-and-go foods and meals during the pandemic, no participants in this study reported eating meals or receiving food from these sites during the recalls. This may have resulted in lower diet quality for adolescents who previously utilized the school meals program for one or more meals per day. Additionally, during the pandemic, all households received the maximum allotment of SNAP benefits [51]. For example, a family of four received $835 during the pandemic, regardless of how much they were receiving prior to the pandemic. Our sample was restricted to households eligible to receive SNAP benefits. Therefore, it may be possible that the lack of differences observed between adolescents with food security and those with food insecurity is because all households in this study were receiving similar amounts of benefits, which potentially allowed them to purchase healthy foods more consistently and influenced the types of foods available in the house. Studies comparing diet quality among adolescents with food security and those with food insecurity in the general population and in non-pandemic times are warranted.

One unexpected finding in this study was that food-insecure participants had a higher average dietary component score for seafood and plant protein compared to food-secure participants. This finding is inconsistent with previous literature, which showed that US households with food insecurity are less likely to consume seafood and plant proteins compared to households with food security [52]. There were no obvious differences in factors influencing seafood consumption between food-secure and food-insecure participants; in both groups, parents were largely responsible for purchasing and preparing seafood for adolescent participants. However, participants with food insecurity were significantly more likely than those with food security to consume peanut butter, which was the only source of plant protein consumed by any participants. Peanut butter is a relatively inexpensive and a convenient option for adolescents who may not have a wide variety of other foods available at home or who may have limited cooking abilities. Peanut butter is also included in the package for WIC participants [53]. Although we did not collect information about household WIC eligibility or participation in this study, it is possible that adolescents experiencing food insecurity lived in WIC-participating households and had more consistent access to peanut butter than those with food security.

### Strengths and Limitations

This study provided information on the overall HEI score and HEI dietary components, which provide a more comprehensive measure of diet than some previous studies (e.g., those that examined only fruit and vegetable consumption or only consumption of added sugar, sweets, snacks, and sugar-sweetened beverages). This study also used innovative methods to assess factors influencing diet quality among adolescents. Although the ASA24 captures some factors, such as time of eating event and activities while eating, this study explored additional factors such as food source, acquisition, and preparation. The addition of these qualitative data provided some explanation as to why there were no differences in overall diet quality between adolescents with food security and those with food insecurity in this sample. Additionally, the sample in this study was largely homogenous, which may allow us to draw conclusions and inform policy and programs for low-income, Black adolescents living in Baltimore City. In Baltimore City, 62% of residents identify as Black or African American, and one-quarter of Black residents live below the federal poverty line [54].

However, the population from which we sampled was not representative of the entire population of adolescents in Baltimore, nor of locations outside of Baltimore. All adolescents in the sample were in households that were eligible to receive SNAP benefits. Because of the nature of the sampling frame, the participants we reached had access to basic technology (e.g., phone) and were able to gain permission from a parent or guardian. None of the adolescents included in this study were in foster care or emancipated from their families, and the implications of this study will not be applicable to adolescents living in unstable or extreme situations. We also do not have information on the employment, education level, English-speaking proficiency, or mental/physical health status of parents or other adults in the household. These factors could impact food availability in the household or present other unique challenges that were not mentioned or explored in this study. Additionally, this study had a relatively small sample size for quantitative analyses and was only powered to detect a ±5-point difference in overall HEI score. Therefore, there may be differences that would only be detected with a larger sample. Food insecurity is measured on a spectrum and—following the validated measurement tool—all adolescents who answered affirmatively to two or more of the nine questions were classified as food insecure. In this study we did not stratify food-secure participants into ‘moderate’ or ‘severe’ categories due to sample size limitations, and there may be differences in overall diet quality or its components by food security severity that were not assessed in this study. Additionally, food security status was measured one to three months prior to the interviews and 24-h dietary recalls. Given the temporal instability of food insecurity and the lag between its assessment and the dietary assessment, it is possible that there may have been some misclassification of participants at the time of the assessment. Misclassification may have contributed to the lack of significant differences in diet quality and contextual factors observed between adolescents with food security and those with food insecurity.

## 5. Conclusions

Diet quality is an important measure related to health in adolescents, as poor diet quality is associated with poor short-term and long-term physical and psychosocial health outcomes. In this study of Black adolescents ages 14–19 living in Baltimore City, we found overall low diet quality and no differences in diet quality between adolescents with food security and those with food insecurity. The qualitative results of this study support findings in previous studies that food choice among adolescents is dependent on multiple factors. We observed that workplace environments enabled adolescents to eat foods high in refined grains, sugar, and saturated fat. We also found that adolescents were largely influenced by their parents and the home food environment, which may be particularly relevant in times when they are home for prolonged periods (i.e., emergency school closures, summer, and winter breaks). Programs and policies that aim to improve healthy food access may positively impact adolescent food security and diet quality. In general, it is important to ensure that healthy foods are available and accessible to adolescents in the places where they spend the most time. Therefore, multilevel interventions in the home, school, and workplace may be most effective in encouraging healthy eating behaviors among adolescents.

## Figures and Tables

**Table 1 nutrients-14-04573-t001:** Demographic characteristics of adolescents aged 14–19 living in Baltimore by food security status (*n* = 61).

Variable	Overall (*n* = 61)	Food-Secure (*n* = 31)	Food Insecure (*n* = 30)
	*n* (%) or Median (IQR)		
**Age**	16 (2.5)	16 (2.5)	16 (2.3)
**Gender**			
Boy	30 (49.2%)	15 (48.3%)	15 (50.0%)
Girl	31 (50.8%)	16 (51.7%)	15 (50.0%)
**Race**			
Black	59 (96.7%)	30 (96.8%)	29 (96.7%)
Biracial (Black/White)	2 (3.3%)	1 (3.2%)	1 (3.3%)
**Youth employment**			
No job	40 (65.6%)	20 (64.5%)	20 (66.7%)
Part-time	19 (31.2%)	10 (32.3%)	9 (30.0%)
Full-time	2 (3.4%)	1 (3.2%)	1 (3.3%)
**Experiencing housing instability**	3 (4.9%)	1 (3.2%)	2 (6.7%)
**HH size**			
1–4 people	27 (45.0%)	15 (48.4%)	12 (41.4%)
>4 people	33 (55.0%)	16 (51.6%)	17 (58.6%)
**Lives in single-parent household**	33 (52.4%)	18 (58.1%)	15 (50.0%)

**Table 2 nutrients-14-04573-t002:** Median Healthy Eating Index 2015 component and total scores, expressed as absolute scores for adolescents ages 14 to 19 years with food security or food insecurity living in Baltimore City (*n* = 61).

Component (Max Score)	Overall (*n* = 61)		Food Secure (*n* = 31)		Food Insecure (*n* = 30)		Difference in Median (FS-FI)	*p*-Value *
	Median	Q1, Q3 ^§^	Median	Q1, Q3 ^†^	Median	Q1, Q3 ^§^		
Total fruits (5)	1.58	0.39, 4.29	1.36	0.16, 4.35	1.74	0.59, 3.87	−0.38	0.90
Whole fruits (5)	0.11	0, 3.20	0.31	0, 2.53	0.10	0, 3.40	+0.21	0.90
Total vegetables (5)	2.47	1.71, 3.33	2.73	1.65, 3.54	2.16	1.72, 2.94	+0.57	0.46
Beans and greens (5)	0.55	0, 3.37	0.2	0, 4.23	0.56	0, 2.16	−0.36	0.56
Whole grains (10)	0.1	0, 1.55	0	0, 1.84	0.59	0, 1.45	−0.59	0.64
Dairy (10)	4.82	3.49, 6.76	5.61	3.69, 6.99	4.09	2.96, 6.40	+1.52	0.17
Total protein foods (5)	5.0	4.51, 5.0	5	4.75, 5	5	4.43, 5.0	0	0.58
Seafood & plant proteins (5)	1.68	0.07, 4.93	0.52	0.02, 2.72	3.23	0.65, 5.0	−2.71	0.02
Fatty acids (10)	3.82	2.35, 8.06	3.82	2.08, 7.88	4.45	2.62, 7.98	−0.63	0.68
Refined grains (10) ^†^	6.97	4.17, 9.15	6.97	5.47, 9.89	7.15	3.56, 8.15	−0.18	0.25
Sodium (10) ^†^	3.17	1.27, 5.10	3.08	0.16, 5.55	3.18	2.35, 4.65	−0.10	0.80
Added sugars (10) ^†^	7.16	5.08, 8.66	7.27	4.78, 8.35	6.65	5.20, 8.99	+0.62	0.87
Saturated fats (10) ^†^	4.42	2.04, 6.16	4.96	2.07, 6.03	4.10	2.07, 6.47	+0.86	0.96
Total HEI score (100)	46.53	37.64, 53.23	46.82	39.28, 53.68	45.82	37.48, 53.09	+1	0.77

Total Healthy Eating Index score has a maximum of 100 points. Total fruits, whole fruits, total vegetables, total protein foods, and seafood and plant proteins have a maximum of 5 points. Whole grains, dairy, fatty acids, refined grains, sodium, added sugars, and saturated fats have a maximum of 10 points. ^†^ Higher scores are given for lower consumption of moderation components (refined grains, added sugars, saturated fats, sodium) and for greater consumption of adequacy components (total fruits, whole fruits, total vegetables, greens and beans, whole grains, dairy, total protein foods, seafood and plant proteins, and fatty acids). ^§^ Quartile 1 (Q1) is the median of the lower half of the data and Quartile 3 (Q3) is the median of the upper half of the data.* *p*-value calculated between food secure and food insecure by the nonparametric Kruskal–Wallis test due to non-normality of data in the distributions of scores for each food group and overall HEI score. Bold indicates *p*-values < 0.05. Abbreviations: Healthy Eating Index (HEI), Food secure (FS), Food insecure (FI).

## Data Availability

Data may be available upon request.

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
