# Peer review of "Diet Quality and Contextual Factors Influencing Food Choice among Adolescents with Food Security and Food Insecurity in Baltimore City"

_nutrients, 2022, doi:10.3390/nu14214573_

Round 1

Reviewer 1 Report

The study examined aspects of the diet and food habits in a black adolescent population and is relevant to concerns about the nutritional status of underprivileged city dwellers. Authors used a combination of valid qualitative and quantitative methods to assess the subjects' food security status and quality of diet. Overall, quality of the paper is good. 

Authors should explain why participants' body-weight status was non assessed. Results would be more meaningful, had this parameter been taken into consideration.

In addition, the following points require clarification or improvement:

Food/nutrient groups under Healthy Eating Scoring include fatty acids as one of its "adequacy components", while saturated fat is grouped with "moderation components". Authors should provide in Methods the  rationale behind total(?) fatty acids (a measure of fat intake) being considered an "adequacy component".

It is mentioned that peanut butter was the only source of plant protein consumed by any participant. Please add which other foods qualify as sources of plant protein, for instance, aren't beans and other pulses considered a source of plant protein?

One of the reasons put forward to explain the study's "unexpected" finding, i.e. that diet quality did not differ by food security, sounds as fallacy (lines 454-458). Here the authors imply that the food-security scale could be more efficient if it "measured factors related to diet  dietary quality"; but this way, an undesirable collinearity effect would be produced.

Reviewer 2 Report

Review of “Diet Quality and Contextual Factors Influencing Food Choice among Adolescents with Food Security and Food Insecurity in Baltimore City” (nutrients-1965001)

This study investigated the differences in diet qualities between adolescents with food security and those with food insecurity. The concept of this study was interesting. However, several problems to be solved.

1.     The introduction section was too long and difficult to follow.

2.     This reviewer does not understand the significance of Figure 1.

3.     The text in the results section was also too long and difficult to follow.

4.     Figures and tables should be used more efficiently.

Round 2

Reviewer 2 Report

The content of Figure 1. HEI index is based on the data of the diet intake; therefore, there was a close association between HEI index and each component. From this point, this reviewer though that Figure 1 was not necessary. This point was revised well. On the other hand, this Figure 1 was based on all the participants. To meet the purpose of this study, it would be better to evaluate food secure and food insecure separately.

Author Response

We appreciate the time Reviewer 2 took to fully explain their point about Figure 1. After further consideration, we have decided to remove Figure 1 and associated text from the manuscript. We agree with Reviewer 2 that the pertinent data relating to our research aims may be found in Table 2.

There were a few places throughout the manuscript that we edited to align with the removal of Figure 1:

a) We reverted our aims back to what they were originally, so they now read: "... the aims of this study were to evaluate the association of food security status with overall diet quality and its components and to examine factors influencing food choice for adolescents with food security and those with food insecurity." (lines 112-114)

b) We added a short description of diet quality for all adolescents in section 3.2 using information from Table 1 (lines 265-273). This provides a complete picture of diet quality for the entire sample population before explaining the results by food security status.

c) We removed one sentence from results section 3.3 that previously referred to the results from Figure 1 (highlighted at line 305 to indicate removal).

d) We removed one sentence from the discussion section that previously referred to the results from Figure 1 (highlighted at line 430 to indicate removal). We did not add any text here because we felt the next sentence, starting with "The qualitative results..." was sufficient.